# Prospective estimation of the age of initiation of cigarettes among young adults (18–24 years old): Findings from the Population Assessment of Tobacco and Health (PATH) waves 1–4 (2013–2017)

Adriana Pérez[1,2]*, Arnold E. Kuk[2], Meagan A. Bluestein[2], Elena Penedo[2,3], Roi San N'hpang[2,3], Baojiang Chen[1,2], Cheryl L. Perry[4], Kymberly L. Sterling[5], Melissa B. Harrell[2,3]

1 Department of Biostatistics and Data Science, School of Public Health, The University of Texas Health Science Center at Houston (UTHealth), Austin, Texas, United States of America, 2 Michael & Susan Dell Center for Healthy Living, School of Public Health, The University of Texas Health Science Center at Houston (UTHealth), Austin, Texas, United States of America, 3 Department of Epidemiology, Human Genetics and Environmental Sciences, School of Public Health, The University of Texas Health Science Center at Houston (UTHealth), Austin, Texas, United States of America, 4 Department of Health Promotion and Behavioral Sciences, School of Public Health, The University of Texas Health Science Center at Houston (UTHealth), Austin, Texas, United States of America, 5 Department of Health Promotion and Behavioral Sciences, School of Public Health, The University of Texas Health Science Center at Houston (UTHealth), Dallas, Texas, United State of America

* Adriana.perez@uth.tmc.edu

## Abstract

### Objectives

To prospectively estimate the age of cigarette initiation among young adults (18–24 years old) who were never cigarette users at their first wave of adult study participation overall, by sex, and by race/ethnicity given recent increases in cigarette initiation occurring in young adulthood.

### Methods

Secondary analyses were conducted using the PATH restricted-use adult datasets among young adult never users of cigarettes in waves 1–3 (2013–2016) with outcomes followed-up in waves 2–4 (2014–2017). Interval censoring survival methods were used to estimate the age of initiation of (i) ever, (ii) past 30-day, and (iii) fairly regular cigarette use. Among never cigarette users when they first entered the adult study, interval censoring Cox proportional hazard models were used to explore differences in the estimated age of initiation of the three cigarette use outcomes by sex and by race/ethnicity, controlling for the effect of previous e-cigarette use and the total number of other tobacco products ever used (0–5 products) before cigarette initiation outcomes.

**Data Availability Statement:** Data Availability Statement: All the data from waves 1-4 are

available from the Population Assessment of Tobacco and Health (PATH) Study [United States] Restricted-Use Files. Inter-university Consortium for Political and Social Research [distributor], 2020-06-24. https://doi.org/10.3886/ICPSR36231. v25. Data are available from https://www.icpsr. umich.edu/web/NAHDAP/studies/36231. This information is listed inside the manuscript as references. Because the PATH restricted datasets requires that we provide the dates of disclosure of the result, each table includes this reference and the dates of disclosure to comply with PATH requirements to present the results in this manuscript.

**Funding:** Research reported in this publication was supported by grant number [R01CA234205] from the National Cancer Institute (NCI) and the FDA Center for Tobacco Products (CTP) to Dr. Adriana Pérez. The content is solely the responsibility of the authors and does not necessarily represent the official views of the National Institutes of Health (NIH) or the Food and Drug Administration (FDA).

**Competing interests:** The authors have no conflicts of interest to disclose except Dr. Harrell is a consultant in litigation involving the vaping industry. This does not alter our adherence to PLOS ONE policies on sharing data and materials.

## Results

Among the young adults who were never cigarette users at their first wave of adult participation, the highest increase in cigarette initiation occurred between 18 and 19 years old. By age 21, 10.6% (95% CI: 9.5–11.7) initiated ever cigarette use, 7.7% (95% CI: 6.1–8.1) initiated past 30-day of cigarette use, and 1.9% (95% CI: 1.4–2.5) initiated fairly regular cigarette use. After controlling for other tobacco products: (a) males were 83% more likely to initiate past 30-day cigarette use at earlier ages than females; (b) Hispanic and Non-Hispanic Black young adults had increased risk to initiate past 30-day cigarette use at earlier ages than Non-Hispanic White young adults (62% and 34%, respectively).

## Conclusions

The substantial amount of cigarette initiation among young adults reinforces the need for prevention strategies among this population. Although, interventions are needed for all young adult populations, strategies should target 18-21-year-olds, with potentially differential prevention targets by sex and by race/ethnicity.

## Introduction

Though the prevalence of combustible tobacco cigarette use has declined since the mid-1960s among adults, cigarette use remains a leading cause of many preventable diseases (i.e., cardiovascular disease, emphysema, chronic bronchitis, and lung cancer, among others) and death in the United States [1–5]. The 2012 Surgeon General's Report found that among 30–39 year olds who had ever tried cigarettes, almost all cigarette initiation happens before the conclusion of early adulthood, with 98% recalling that their first cigarette use occurred before 26 years of age [6]. Among young adults ages 18–24 in the 2013–2014 Population Assessment of Tobacco and Health (PATH) study, the prevalence of ever cigarette use was 53.2% and the prevalence of past 30-day cigarette use was 25.5% [7]. The same PATH study found that the prevalence of past 30-day cigarette use was higher in young adults than in adults 25 years old and older [7]. The National Survey on Drug Use and Health (NSDUH) from 2002–2015 examined the cross-sectional incidence of cigarette initiation among youth (ages 12–17) and young adults (ages 18–21 and 22–25), which found that while initiation of cigarette use was declining among all these age groups, cigarette initiation remained the highest among the 18–21 year olds (13% vs. 4% in 22–25 year olds) [8]. A recent study of the cross-sectional NSDUH 2002–2018 found that the recalled age of cigarette initiation occurring in young adulthood (18–23 years old) increased from 20.6% in 2002 to 42.6% in 2018 [9], indicating that young adults are increasingly vulnerable to initiating cigarette use.

A longitudinal study of cigarette initiation among PATH young adults (18–24 years old) who had never used cigarettes in 2013–2014 found after 1 year of follow-up that 6.8% initiated ever cigarette use and 4.5% had initiated past 30-day cigarette use, which was higher than the proportions of initiation for youth ages 12–17 years old (ever use = 3.8% and past 30-day use = 1.6%) [10]. Another longitudinal study among PATH young adults (18–24 years old) that included never cigarette users from both 2013–2014 and 2014–2015 found after 1 year of follow-up that 6.0% had initiated ever cigarette use and 3.6% had initiated past 30-day cigarette use. This was higher than the proportions of cigarette initiation among youth 12–17 years old (ever use = 4.4% and past 30-day use = 2.0%) [11]. In addition, a previous report of PATH

among young adults reported initiation of ever (10.3%) and past 30-day (6.1%) cigarette use after 1 or 2 years of follow-up [12]. Taken together, these findings show that young adults need to be thoroughly examined given that cigarette initiation still occurs during this period.

Other studies also showed that certain racial/ethnic groups report increased cigarette initiation in young adulthood rather than adolescence [13–15]. Previous reports found that larger percentages of Asian/Pacific Islander, African American, and Hispanic young adults report recalling initiating cigarette use in young adulthood compared to Non-Hispanic white young adults [13]. NSDUH data from 2006–2008 indicated that while Non-Hispanic white young adults (ages 18–25) had the highest proportions of cigarette initiation, this trend reverses at ages 26–34, with African American young adults reporting the highest prevalence of cigarette initiation in comparison to Non-Hispanic white young adults during this period [14].

The age of cigarette initiation is an important factor to study because age of initiation is associated with the transition to daily smoking [16], nicotine dependence [6], risk of lung cancer and other smoking-related diseases, including differences by sex and race/ethnicity [17–19]. In all of the aforementioned PATH studies, while cigarette use prevalence and incidence has been examined, the *age* of cigarette initiation was not estimated. Given that cigarette use has declined in recent years [1,6], there is still a substantial amount of initiation occurring in young adulthood [8,10,11], some young adults (i.e., those aged 21 and older) are the legal targets of the tobacco industry [20], and that the tobacco marketplace has diversified with the introduction of e-cigarettes and other tobacco products [21], we prospectively measured age of cigarette initiation in young adults in a USA nationally representative cohort. In this study we examined the age of three cigarette initiation outcomes (ever, past 30-day, and fairly regular use) in young adults (18–24 years old) across 4 waves (2013–2017) of the PATH study overall, by sex, and by race/ethnicity, while controlling for previous e-cigarette use and the total number of other tobacco products ever used before cigarette initiation because of the relationship between previous e-cigarette use and subsequent combustible cigarette initiation [11,22].

## Methods

The PATH study is an ongoing nationally representative longitudinal cohort study of U.S. youth and adults conducted annually since 2013 that assesses tobacco use behaviors, attitudes, beliefs and tobacco-related health outcomes [23]. The original investigators of the PATH study obtained informed consent for participants 18 and older and parental consent for youth 12–17 years old [23] providing written assent and each youth's parent/legal guardian providing written consent. In addition, all data were de-identified in order to preserve participant anonymity [23]. Further details about PATH and its sampling methodology are described elsewhere [23]. The wave 1 young adult (18–24 years old) sample included 9,110 participants [24]. IRB approval for this study was obtained from the Committee for the Protection of Human Subjects at the University of Texas Health Science Center at Houston with number HSC-SPH-17-0368.

### Study design and participants

This study conducted secondary analyses using the PATH restricted-use adult datasets among young adult (18–24 years old) never users of cigarettes in waves 1–3 (wave 1: 2013–2014, wave 2: 2014–2015, wave 3: 2015–2016) with outcomes followed-up in waves 2–4 (wave 2: 2014–2015, wave 3: 2015–2016, wave 4: 2016–2017). Among the 9,110 wave 1 participants, 3,135 were never users of cigarettes. Additionally, we included "aged-up" youth (i.e., those who participated in the PATH youth survey when they were 17 years old or 16 years old at wave 1 and subsequently participated in the adult survey when they turn 18 in waves 2 or 3) who had

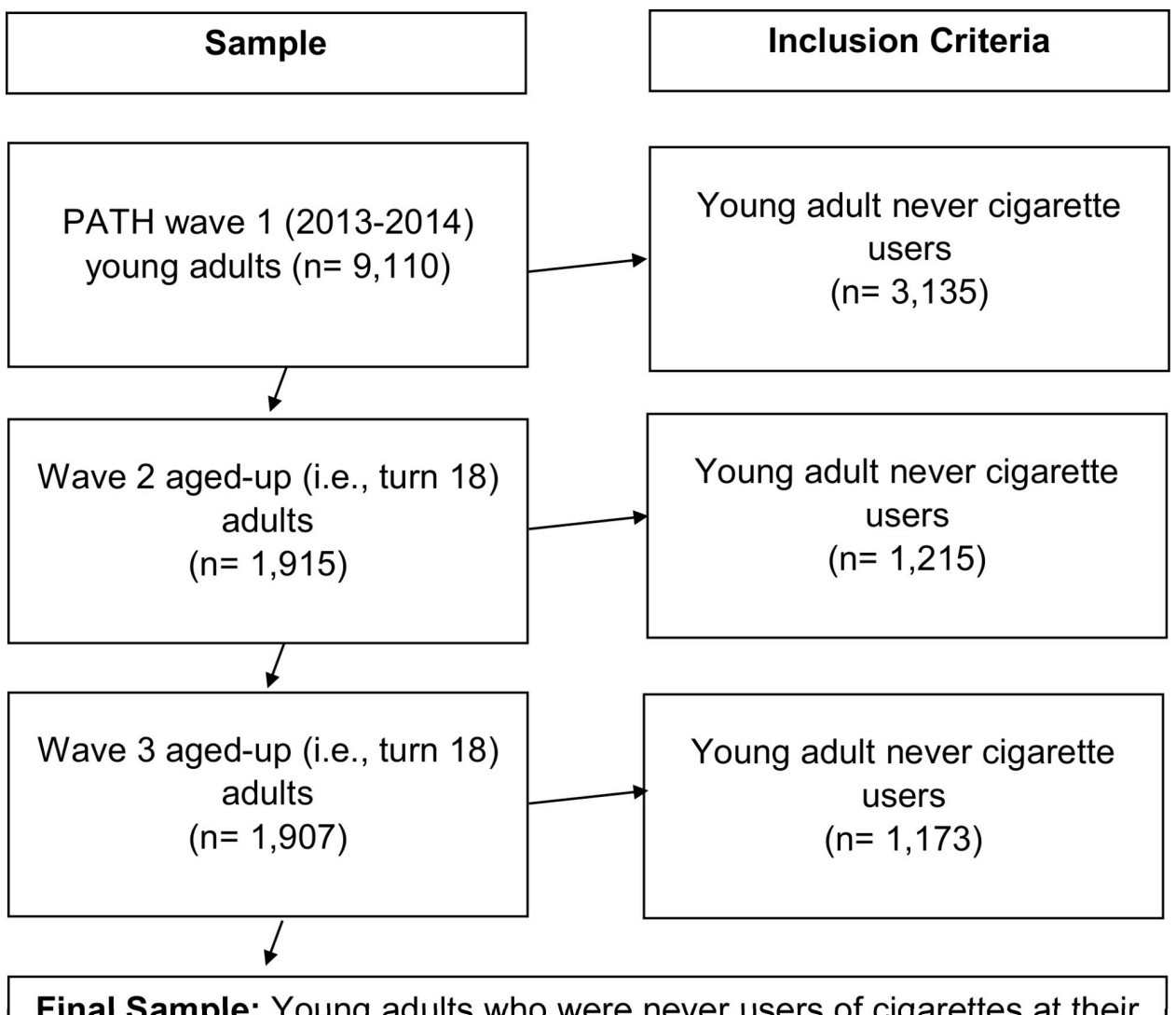

**Fig 1. Data management and participant eligibility.**

never used cigarettes at their first wave of PATH adult participation (i.e., when they turned 18). This resulted in 1,215 participants being included in our study at 2014–2015 and 1,173 participants being included in our study at 2015–2016 [25]. New cohort participants who entered the wave 4 PATH study in 2016–2017 and aged-up participants in 2016–2017 are not used since they do not have a follow-up period for the cigarette initiation outcomes, as the PATH data for the following years had not been released at the time of writing this manuscript. In total, along with the never cigarette users from 2013–2014, this resulted in 5,523 (weighted frequency, or N = 19,548,811) young adult never cigarette users at their first wave of adult study participation being included in the analysis. Participants who were 18–24 years old at wave 1 and were followed-up through wave 4 were a maximum of 27 years old. Fig 1 displays

the flow of data management and eligibility of participants that was used to derive the final sample.

## Cigarette use measures

The three outcomes analyzed include the age of initiation of: ever cigarette use, past 30-day cigarette use, and fairly regular cigarette use. The PATH study at waves 2–4 asks participants: "Have you ever smoked a cigarette, even one or two puffs?". Response options were "yes", "no", "I don't know", and "refused". Past 30-day cigarette use was measured in waves 2–4 with the question, "In the past 30 days, on how many days did you smoke cigarettes?". Numeric response options included 0–30 days and participants were considered past 30-day cigarette users if they reported cigarette use on 1 or more days. Fairly regular cigarette use was measured in waves 2–4 in the adult questionnaire by "Have you ever smoked cigarettes fairly regularly?". Response options included "yes", "no", "I don't know", and "refused". Participants who answered "I don't know" or refused to answer each question were excluded from analysis for that outcome.

## Previous E-cigarette and other tobacco product use

Similar variables to represent ever use for the following other tobacco products were available in the restricted datasets: e-cigarettes, traditional cigars, cigarillos, filtered cigars, hookah and smokeless tobacco. Participants who reported "yes" for each product's ever use were considered ever users. Those who answered "no" were considered never users for each product and those who reported "I don't know" or "refused" were excluded from analysis. Given previous publications on the association between e-cigarette use and subsequent cigarette use [11,22], we decided to control for the potential association between previous ever e-cigarette use (yes/no) and each age of cigarette initiation outcome, as well as the total number of other tobacco products ever used. For this purpose, we created three variables identifying ever e-cigarette use prior to initiation of each cigarette use outcome. In addition, we created three variables representing the total number of five other tobacco products ever used prior to initiation of the three cigarette use outcomes (traditional cigars, cigarillos, filtered cigars, hookah and smokeless tobacco). The sum of the total number of other tobacco products ever used before cigarette initiation was treated continuously (0–5 tobacco products).

## Demographic variables

Sex was classified by PATH as males or females. The sex variable was imputed by PATH at wave 1 but not at waves 2 and 3. The following four categories were used to measure participant's race in PATH: White race alone, Black race alone, Asian race alone, and other race (including multi-racial). Participants' ethnicity was categorized as either Hispanic or Non-Hispanic. In order to have comparable data with the Surgeon General's reports [1,6,21], race and ethnicity were combined to create the following four race/ethnicity categories for young adults: Non-Hispanic White, Hispanic, Non-Hispanic Black, and Non-Hispanic Other (including Non-Hispanic Asian, multi-race, and other races).

## Age of initiation

PATH provided a derived variable for participant age in years at each wave (waves 1–4) because date of birth is not provided to researchers in the restricted datasets [26,27]. PATH provided an additional derived variable to represent the number of weeks between waves of study participation that is calculated by assigning the calendar week of the year (0–52) to each

date that the survey was conducted. It should be noted that PATH survey waves were administered approximately once per year, but the number of weeks between survey waves could have exceeded 52 weeks for some participants. Age of initiation of ever cigarette use, past 30-day cigarette use, and fairly regular use was estimated by adding the participants' age at the first wave, 2013–2016 (waves 1–3), in which they were never cigarette users to the number of weeks between relevant subsequent waves, 2014–2017 (waves 2–4), in which the participants first reported initiating each cigarette outcome. Participant age was calculated in weeks and added to the number of weeks between waves of participation to provide a more precise estimate of participant age. In waves 2–4, a lower and an upper age bound for each cigarette outcome was calculated. The lower age bound was determined by using the age at the last wave participants did not initiate each cigarette outcome. For participants who initiated each cigarette outcome, the upper age bound was the age at the first wave where they reported initiating each cigarette outcome. Participants who did not initiate each outcome through the study follow-up period were considered censored in their upper age bound. Finally, the lower and upper age bounds were converted from age in weeks back to age in years on a continuous scale.

## Statistical analysis and data management

Data analyses incorporated sampling weights and the 100-balance repeated replicate (BRR) weights to account for PATH's complex study design with Fay's adjustment set to 0.3 to increase estimate stability [23,26–28]. Weighted means for continuous variables and weighted proportions for categorical variables are provided. Weighted non-parametric interval censoring methods for time-to-event analyses were implemented to estimate the probability of the age of initiation of ever cigarette use, past 30-day cigarette use and fairly regular cigarette use [29–31]. The hazard function for each outcome was estimated overall, by sex, and by race/ethnicity using the Turnbull non-parametric estimator [32], reported as cumulative incidence in percentages and are presented in figures [33]. This resulted in seven interval-censored hazard functions for each outcome: overall, males, females, Non-Hispanic White, Hispanic, Non-Hispanic Black, and Non-Hispanic Other. We did not stratify the hazard functions by previous e-cigarette use or the total number of other tobacco products ever used because we used these variables to evaluate if any effects by sex and by race/ethnicity hold after controlling for these variables. Differences in the age of cigarette initiation for each of the three outcomes by sex, by race/ethnicity, while controlling for previous e-cigarette use and the total number of other tobacco products ever used were estimated by fitting weighted Cox proportional hazard regression models to interval-censored data with a piecewise constant function as the baseline hazard function [34]. Crude hazard ratios and 95% CIs are reported. Multivariable Cox models including sex and race/ethnicity, while controlling for previous e-cigarette use and the total number of other tobacco products ever used were fit for each outcome to provide adjusted hazard ratios (AHRs) and 95% CIs. All statistical analyses were completed in SAS version 9.4 [35] using the Inter-university Consortium for Political and Social Research server hosted by the University of Michigan.

## Results

Table 1 provides demographic characteristics of young adults (18–24 years old) who were never cigarette users at their first wave of PATH adult participation (waves 1–3, 2013–2016). Among these young adults, 73.1% entered the PATH study at wave 1 (2013–2014). Their average age at their first wave of adult participation was 20 years old; 53.8% were female, and 51.7% were Non-Hispanic white young adults. Additionally, 16.1% of the young adults reported they ever used e-cigarettes before initiation of ever cigarette use, 17.0% before

**Table 1. Demographic characteristics of PATH young adult (ages 18–24 years old) never cigarette users at their first wave of adult study participation, 2013–2016\*.**

| | | Never cigarette users at first wave of adult study participation | |
|---|---|---|---|
| | | n = 5,523; N = 19,548,811[a] | |
| | | Unweighted n (N) | Weighted % (SE) |
| **Wave of entry into study** | Wave 1 (2013–2014) | 3,135 (14,297,143) | 73.1 (0.46) |
| | Wave 2 (2014–2015) | 1,215 (2,704,043) | 13.8 (0.28) |
| | Wave 3 (2015–2016) | 1,173 (2,547,626) | 13.0 (0.29) |
| **Age at the first wave of adult participation: Weighted mean (SE)** | | 20.02 (0.04) | |
| **Sex** | Male | 2,499 (9,023,884) | 46.2 (0.60) |
| | Female | 3,021 (10,516,717) | 53.8 (0.60) |
| | Missing Values | 3 | |
| **Race/Ethnicity** | Non-Hispanic White | 2,606 (10,099,160) | 51.7 (1.10) |
| | Hispanic | 1,374 (3,867,209) | 19.8 (0.69) |
| | Non-Hispanic Black | 989 (3,089,826) | 15.8 (0.65) |
| | Non-Hispanic Other[b] | 554 (2,492,617) | 12.7 (0.89) |
| **Previous e-cigarette use before cigarette initiation outcomes (SE)** | Ever use | 16.1% (0.65%) | |
| | Past 30-day use | 17.0% (0.69%) | |
| | Fairly regular use | 18.6% (0.74%) | |
| **Average number of other tobacco products ever used before cigarette initiation outcomes (SE)[c]** | Ever use | 5,344 (18,783,451) | 0.45 (0.02) |
| | Past 30-day use | 5,344 (18,777,712) | 0.47 (0.02) |
| | Fairly regular use | 5,349 (18,797,295) | 0.50 (0.02) |

\* PATH restricted file received disclosure to publish: July 23, 2020, February 26, 2021, and March 12, 2021. United States Department of Health and Human Services. National Institutes of Health. National Institute on Drug Abuse, and United States Department of Health and Human Services. Food and Drug Administration. Center for Tobacco Products. Population Assessment of Tobacco and Health (PATH) Study [United States] Restricted-Use Files. ICPSR36231-v13.AnnArbor, MI: Inter-university Consortium for Political and Social Research [distributor], November 5, 2019. https://doi.org/10.3886/ICPSR36231.v13.

a: N = weighted frequency.

b: Non-Hispanic Other include Non-Hispanic Asian, Multi-race, and etc.

c: Weighted frequency distributions of the total number of other tobacco products ever used before ever, past 30-day, and fairly regular cigarette initiation are provided in S1 Table.

initiation of past 30-day cigarette use, and 18.6% before initiation of fairly regular cigarette use. The mean number of other tobacco products ever used before each cigarette initiation outcome are reported in Table 1 and S1 Table provides the weighted proportions for the total number of other tobacco products ever used (0–5) before cigarette initiation outcomes.

Table 2 shows the distribution of the prospectively estimated age of initiation of cigarettes for the three outcomes for the young adults who were never cigarette users at their first wave of adult participation, and Fig 2 displays these results graphically. The highest increase in initiation occurs between 18 and 19 years old for ever and past 30-day cigarette use, specifically an increase of 6.0% (95%CI = 5.3–6.7) for ever use, and 4.0% (95%CI = 3.5–4.6) for past 30-day use. The highest increases in fairly regular cigarette use was observed between 24 and 25 years (from 2.6% to 4.2%). Among the young adult never cigarette users at their first wave of adult participation, by 21 years old, 10.6% had initiated ever use, 5.7% had initiated past 30-day and 1.9% had initiated fairly regular cigarette use. By 25 years old (the upper age limit of young adulthood), 17.0% initiated ever cigarette use, 12.3% initiated past 30-day of cigarette use, and 4.2% initiated fairly regular cigarette use. By 27 years old (the latest age for which we had fol-low-up), 22.4% of young adults who were never cigarette users at first wave of adult participa-tion initiated ever cigarette use and 16.0% initiated past 30-day cigarette use. There were not

**Table 2. Estimated cumulative incidence (and 95% confidence intervals)[a] of the age of initiation of cigarette outcomes for young adult (18–24 years old) never cigarette users at their first wave of PATH adult participation[*].**

| Age | Ever Cigarette Use | Past 30-Day Cigarette Use | Fairly Regular Cigarette Use |
|---|---|---|---|
| **18** | 0.0% | 0.0% | 0.0% |
| **19** | 6.0 (5.3,6.7) | 4.0 (3.5,4.6) | 0.9 (0.3,1.5) |
| **20** | 8.5 (7.4,9.6) | 5.7 (4.5,6.8) | 1.0 (0.3,1.7) |
| **21** | 10.6 (7.3,14.0) | 5.7 (4.9,6.4) | 1.9 (0.9,2.9) |
| **22** | 12.5 (9.7,15.4) | 7.7 (5.5,9.8) | 1.9 (1.4,2.5) |
| **23** | 12.5 (10.6,14.5) | 10.0 (8.1,12.0) | 2.6 (1.9, 3.3) |
| **24** | 17.0 (13.7,20.3) | 11.1 (9.1,13.0) | 2.6 (1.9,3.3) |
| **25** | 17.0 (15.2,18.8) | 12.3 (9.2,15.4) | 4.2 (2.1,6.3) |
| **26** | 18.9 (17.0,20.9) | 13.1 (11.3,15.0) | 4.2 (2.8,5.6) |
| **27** | 22.4 (17.1,27.6) | 16.0 (10.2,21.8) | N/A |

[a]: 95%CI: Turnbull 95% confidence interval.

[*]PATH restricted file received disclosure to publish: May 1 and June 22, 2020. United States Department of Health and Human Services. National Institutes of Health. National Institute on Drug Abuse, and United States Department of Health and Human Services. Food and Drug Administration. Center for Tobacco Products. Population Assessment of Tobacco and Health (PATH) Study [United States] Restricted-Use Files. ICPSR36231-v13.AnnArbor, MI: Inter-university Consortium for Political and Social Research [distributor], November 5, 2019. https://doi.org/10.3886/ICPSR36231.v13.

enough participants who reported initiation of fairly regular cigarette use by 27 years old, but by age 26, 4.2% initiated fairly regular use.

Table 3 shows results from the crude and adjusted interval-censored Cox proportional hazard models evaluating differences in the age of initiation of the three cigarette initiation outcomes by sex and by race/ethnicity, while controlling for previous e-cigarette use, and the total number of other tobacco products ever used before cigarette initiation. Both the crude and adjusted models showed that there were statistically significant differences in the age of initiation by sex, by race/ethnicity, by previous e-cigarette use, and by the total number of other tobacco products ever used in all three cigarette initiation outcomes. The cumulative incidence for each cigarette outcome are displayed graphically in Figs 3 and 4 by sex and by race/ethnicity. After controlling for race/ethnicity, previous e-cigarette use, and the total number of other tobacco products ever used, males had a higher risk of initiating ever cigarette use (AHR: 1.62, 95% CI: 1.33–1.97) and past 30-day cigarette use (AHR: 1.66, 95% CI: 1.27–2.16) at earlier ages compared to females. After controlling for sex, previous e-cigarette use, and the total number of other tobacco products ever used, while Hispanic young adults had a 54% (AHR: 1.54; 95% CI: 1.26, 1.87) higher risk of initiating ever cigarette use at earlier ages compared to Non-Hispanic White young adults, there were no statistically significant differences in the age of initiation between Non-Hispanic Black and Non-Hispanic Other compared to Non-Hispanic White young adults. Hispanic young adults also had a 68% (AHR: 1.68; 95% CI: 1.31, 2.16) higher risk and Non-Hispanic Black young adults had a 41% (AHR: 1.41; 95% CI: 1.07, 1.86) higher risk of initiating past 30-day cigarette use at earlier ages compared to Non-Hispanic White young adults. There were no statistically significant differences in the age of initiation for past 30-day cigarette use between Non-Hispanic Other and Non-Hispanic White young adults. Non-Hispanic Other young adults had a 63% (AHR: 0.37; 95% CI: 0.15, 0.95) lower risk of initiating fairly regular cigarette use at earlier ages compared to Non-Hispanic White young adults. There were no statistically significant differences in the age of initiation of fairly

a) Ever cigarette use

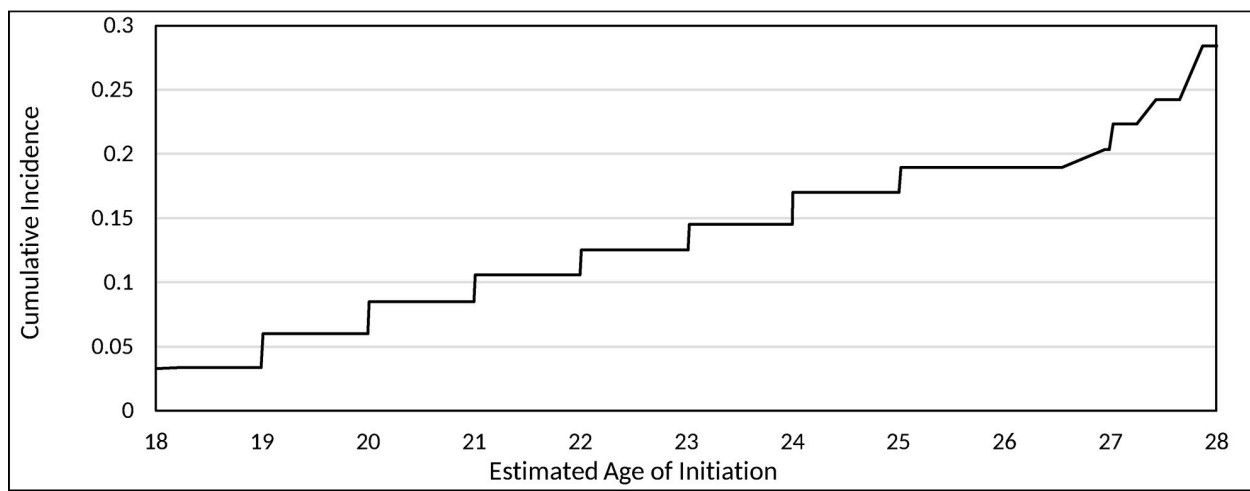

b) Past 30-day cigarette use

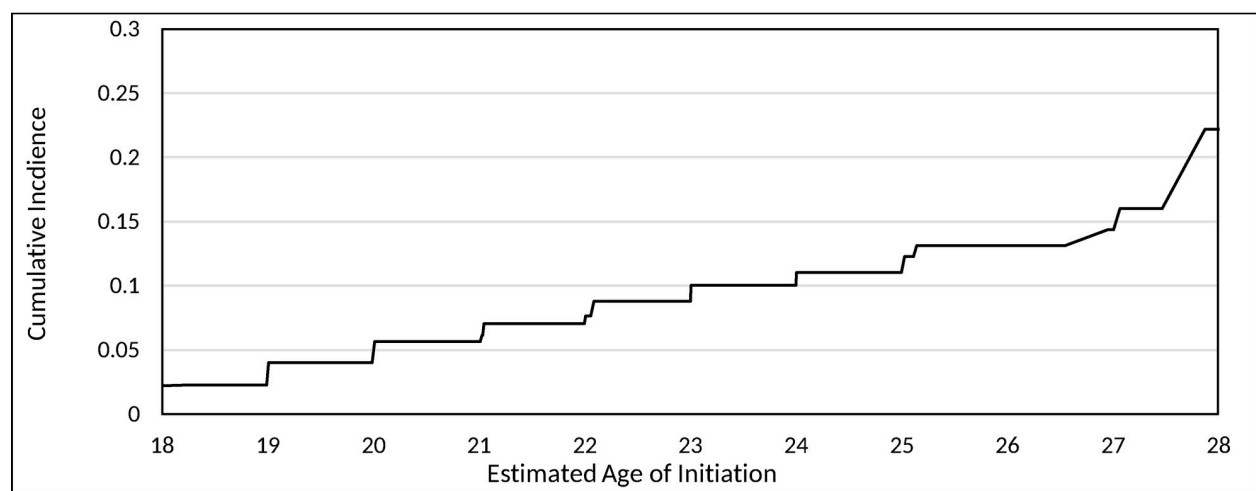

c) Fairly regular cigarette use

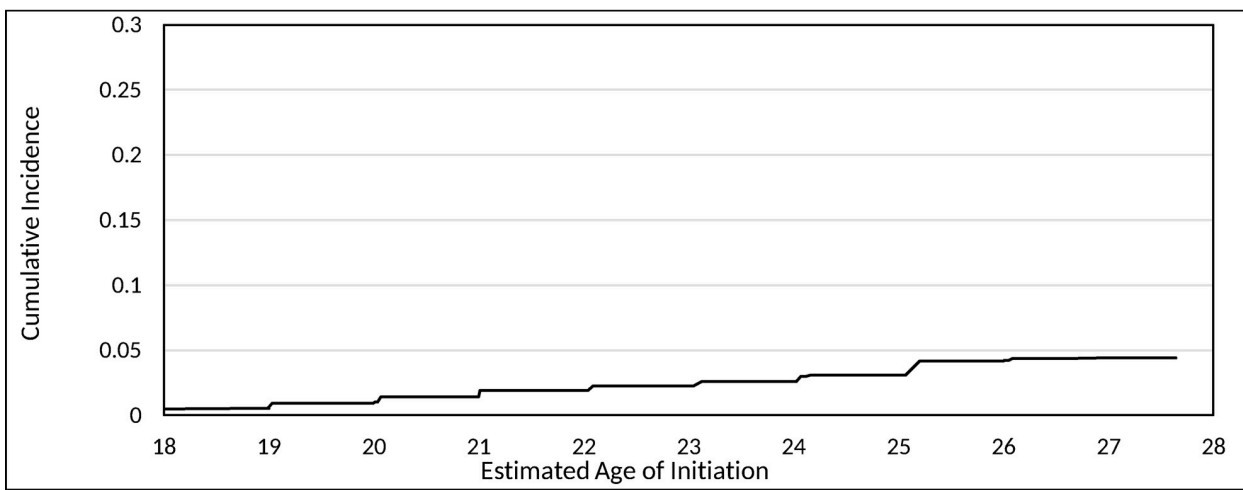

**Fig 2. Estimated cumulative incidence for the age of initiation of cigarette outcomes.**

**Table 3. Crude and adjusted hazard ratios (95% Confidence Intervals) for age of initiation of cigarette outcomes.**

| | Ever Cigarette Use | Past 30-Day Cigarette Use | Fairly Regular Cigarette Use |
|---|---|---|---|
| **Crude hazard ratios** | | | |
| **Sex** | | | |
| Female | 1.00 | 1.00 | 1.00 |
| Male | **1.77 (1.47,2.13)** | **1.83 (1.44,2.33)** | **1.82 (1.13,2.91)** |
| **Race/Ethnicity** | | | |
| Non-Hispanic White | 1.00 | 1.00 | 1.00 |
| Hispanic | **1.53 (1.25,1.87)** | **1.62 (1.26,2.06)** | 1.05 (0.64,1.74) |
| Non-Hispanic Black | 1.05 (0.82,1.35) | **1.34 (1.01,1.78)** | 0.69 (0.35,1.33) |
| Non-Hispanic Other[a] | 0.76 (0.49,1.18) | 0.73 (0.42,1.28) | **0.33 (0.14,0.77)** |
| **Previous e-cigarette use** | | | |
| No | 1.00 | 1.00 | 1.00 |
| Yes | **2.40 (2.04, 2.84)** | **2.31 (1.89, 2.83)** | **2.91 (1.82, 4.67)** |
| Total number of other tobacco products ever used before cigarette initiation | **1.43 (1.34, 1.53)** | **1.42 (1.31, 1.53)** | **1.52 (1.30, 1.76)** |
| **Adjusted Model** | | | |
| **Sex** | | | |
| Female | 1.00 | 1.00 | 1.00 |
| Male | **1.62 (1.33, 1.97)** | **1.66 (1.27, 2.16)** | 1.56 (0.91, 2.68) |
| **Race/Ethnicity** | | | |
| Non-Hispanic White | 1.00 | 1.00 | 1.00 |
| Hispanic | **1.54 (1.26, 1.87)** | **1.68 (1.31, 2.16)** | 0.98 (0.57, 1.70) |
| Non-Hispanic Black | 1.06 (0.84, 1.35) | **1.41 (1.07, 1.86)** | 0.73 (0.37, 1.45) |
| Non-Hispanic Other[a] | 0.84 (0.54, 1.32) | 0.85 (0.48, 1.50) | **0.37 (0.15, 0.95)** |
| **Previous e-cigarette use** | | | |
| No | 1.00 | 1.00 | 1.00 |
| Yes | **1.78 (1.47, 2.16)** | **1.71 (1.32, 2.21)** | **2.19 (1.17, 4.10)** |
| Total number of other tobacco products ever used before ever cigarette initiation | **1.27 (1.17, 1.39)** | **1.26 (1.13, 1.40)** | **1.25 (1.01, 1.57)** |

[a]: Non-Hispanic Other include Non-Hispanic Asian, Multi-race, and etc.

* PATH restricted file received disclosure to publish: October 16, 2020, February 26, 2021, and March 9, 2021. United States Department of Health and Human Services. National Institutes of Health. National Institute on Drug Abuse, and United States Department of Health and Human Services. Food and Drug Administration. Center for Tobacco Products. Population Assessment of Tobacco and Health (PATH) Study [United States] Restricted-Use Files. ICPSR36231-v13.AnnArbor, MI: Inter-university Consortium for Political and Social Research [distributor], November 5, 2019. https://doi.org/10.3886/ICPSR36231.v13.

regular cigarette use between Hispanic and Non-Hispanic Black compared to Non-Hispanic White young adults.

Table 4 shows the estimated cumulative incidence of the age of initiation for each of the cigarette use outcomes. Among never cigarette users at their first wave of adult participation, 13.4% of male, 7.8% of female, 10.1% of Non-Hispanic White, 11.7% of Hispanic, 10.0% of Non-Hispanic Black, and 6.5% of Non-Hispanic Other young adults had initiated ever cigarette use by age 21. Additionally, 7.4% of male, 4.2% of female, 5.4% of Non-Hispanic White, 7.5% of Hispanic, 7.6% of Non-Hispanic Black, and 3.3% of Non-Hispanic Other young adults initiated past 30-day cigarette use by age 21. Among our sample of never cigarette users, we found that 2.7% of male, 1.1% of female, 2.2% of Non-Hispanic White, 2.7% of Hispanic, 1.0% of Non-Hispanic Black, and 0.7% of Non-Hispanic Other young adults initiated fairly regular cigarette use by age 21.

a) Ever cigarette use

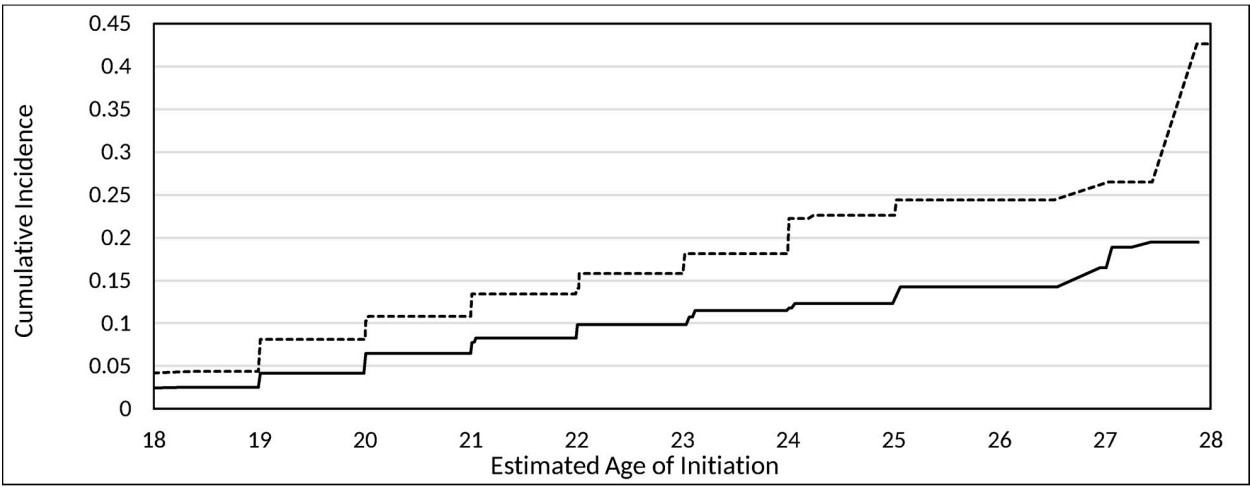

b) Past 30-day cigarette use

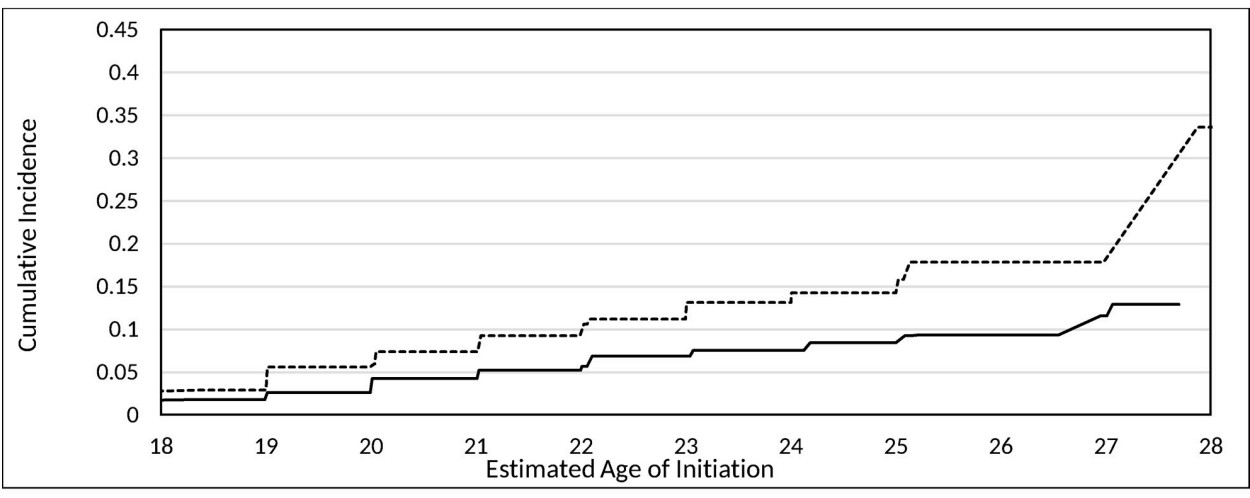

c) Fairly regular cigarette use

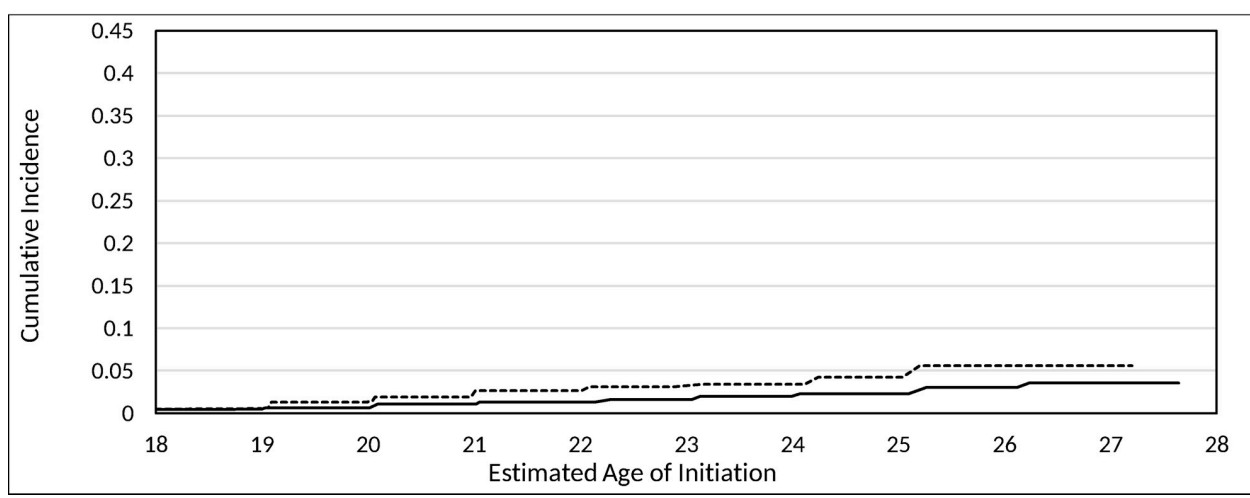

**Fig 3. Estimated cumulative incidence for the age of initiation of cigarette outcomes stratified by sex: Males are the dotted lines and females are the sold line.**

## Discussion

As cigarettes remain one of the most commonly used tobacco products in the US, especially among young adults [9,12,36], it is important to estimate the age of initiation of cigarettes to further inform tobacco regulatory science to add to the body of research on prevention efforts for young adults who did not initiate cigarette use in adolescence. To the best of our knowledge, this study is the first to provide prospective estimates of the age of initiation (2014–2017) for ever cigarette use, past 30-day cigarette use, and fairly regular cigarette use among young adults (age 18–24 years old) who were never cigarette users in the US between 2013–2016. After controlling for previous e-cigarette use and the total number of other tobacco products ever used, we found differences in the age of cigarette initiation by sex and race/ethnicity.

In our study (2013–2017), 17% of young adults who were never cigarette users initiated ever cigarette use by 24 and 22.4% initiated by 27 years old. This is similar to findings from a different study that examined cigarette initiation in young adults who did not initiate cigarette use during adolescence, as that study found that 25% of young adults first tried cigarettes between 18–21 years old [37].

In a different PATH study using data from waves 1–3, researchers assessed past 12-month use, past 30-day use, and frequent past 30-day use of cigarettes (e.g., use 20 or more days in the last 30 days). Among wave 1 young adult (18–24 years old) never users of cigarettes, 10.3% (95% CI: 8.8, 12.0) initiated ever cigarette use, and 6.1% (95%CI = 5.1–7.4) initiated past 30-day cigarette use, and 1.1% initiated frequent past 30-day cigarette use (95%CI = 0.7–1.6) by waves 2 or 3 [12]. One difference between the results of our study and other studies is that we provide the cumulative incidence based on how our study and analyses were conducted, while other studies reported prevalence rates. While previous studies of PATH with shorter follow-up periods have found that among never cigarette users, initiation ranges between 6.0%- 10.3% [10–12], we estimate that 22.4% of those who initiate ever cigarette use do so by age 27 after being followed-up through 2014–2017. Similarly, these studies found that initiation of past 30-day use ranges between 3.6%- 6.1% [10–12], while we estimate that among never cigarette users who initiate past 30-day cigarette use, 16.0% do so by age 27 after being followed-up through 2014–2017. While the results from the 2018 National Health Interview Survey (NHIS) suggested that an estimated 13.7% (34.1 million) of US adults are current cigarette users (has smoked more than 100 cigarettes in their lifetime and smoked every day or some days at the time of the survey) and 7.8% of young adults (18–24) were current users of cigarettes [38], our study estimated more than 3.13 million (19,548,811*0.160) past 30-day cigarette users initiate by the age of 27 between 2013–2017. Furthermore, the 2002–2018 NSDUH reported that cigarette initiation is increasing in young adulthood [9]. Taken together, these findings show that cigarette use is increasing in young adulthood and is still a problem for millions of young adults. We observed significant differences in the age of initiation of ever cigarette use by sex in our study in which males are more likely to initiate ever cigarette use at an earlier age during young adulthood than females, which is consistent with previous studies [1,6,7,37,39]. For example, the Minnesota Adolescent Community Cohort Study, a longitudinal study of cigarette initiation in young adults, found that among never cigarette users at 18 years old, 30% of males vs. 20.4% of females initiated cigarette use between 18–21 years old [37]. Our study reports a similar sex difference, specifically that among never cigarette users, 13.4% of males and 8.3% of females initiated ever cigarette use by age 21 [37]. Our study goes beyond previous reports of sex differences in cigarette initiation by finding that males have

a) Ever cigarette use

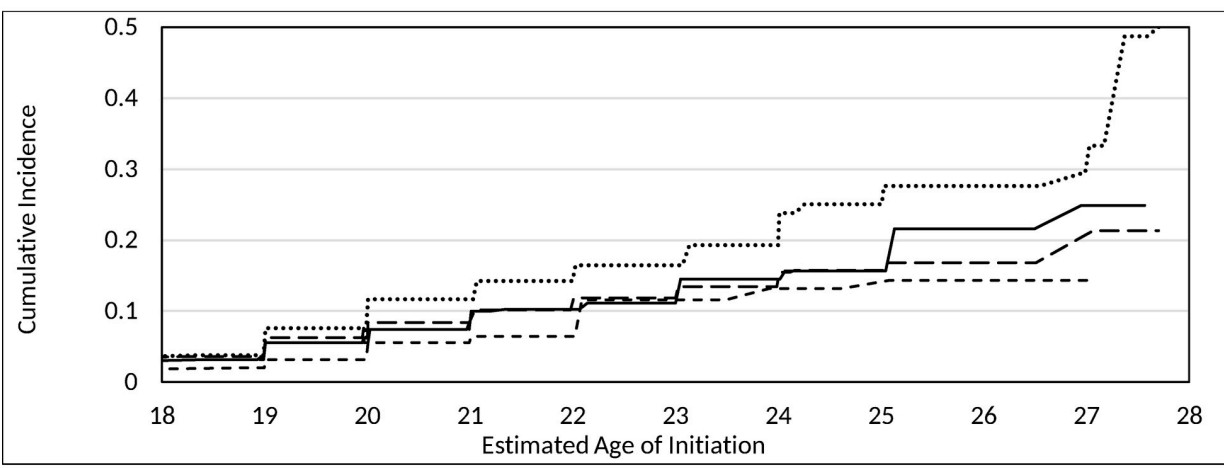

b) Past 30-day cigarette use

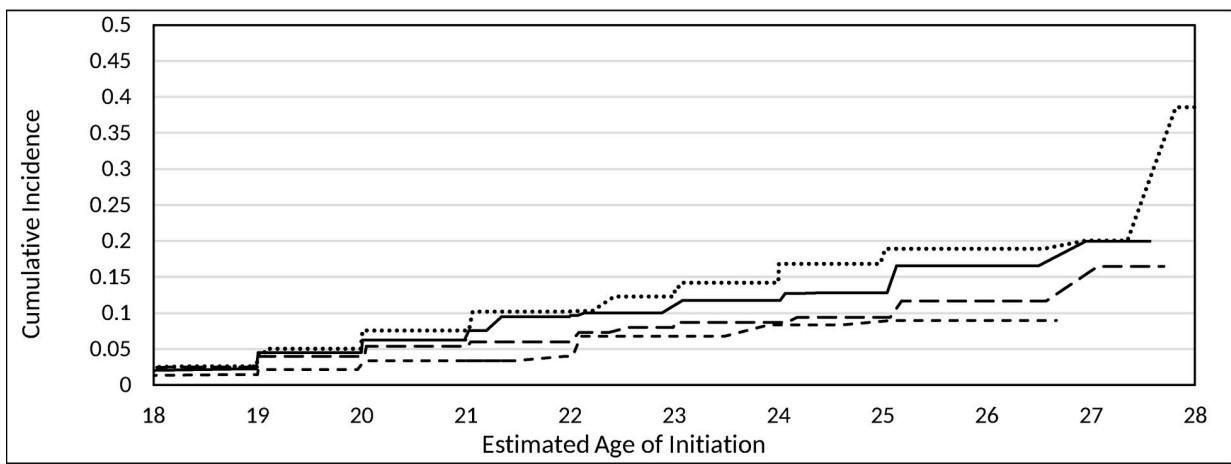

c) Fairly regular cigarette use

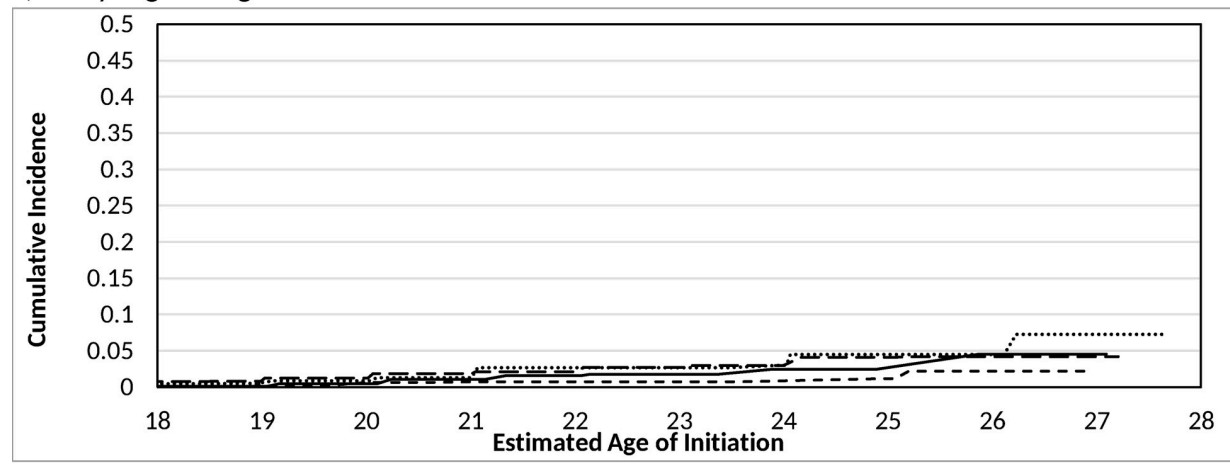

......... Hispanic  — — - Non-Hispanic White  - - - - Non-Hispanic Others  ——— Non-Hispanic Black

**Fig 4. Estimated cumulative incidence for the age of initiation of cigarette outcomes stratified by race/ethnicity.**

**Table 4. Estimated cumulative incidence (and 95% CIs) of the age of initiation of different cigarette outcomes by sex and by race/ethnicity[*].**

| Age | Sex | | Race/Ethnicity | | | |
|---|---|---|---|---|---|---|
| | Males | Females | Non-Hispanic White | Hispanic | Non-Hispanic Black | Non-Hispanic Other[a] |
| **Initiation of ever cigarette use** | | | | | | |
| 18 | 0.0% | 0.0% | 0.0% | 0.0% | 0.0% | 0.0% |
| 19 | 8.1 (6.8,9.5) | 4.2 (3.5,4.8) | 6.2 (5.2,7.3) | 7.6 (6.1,9.2) | 5.5 (3.3, 7.8) | 3.1 (1.1, 5.2) |
| 20 | 10.3 (6.2,14.3) | 6.5 (5.5,7.5) | 8.4 (5.2,11.6) | 11.7 (8.5,14.8) | 5.5 (2.8, 8.3) | 5.5 (0.9,10.2) |
| 21 | 13.4 (8.6,18.3) | 7.8 (5.0,10.6) | 10.1 (8.5,11.8) | 11.7 (8.1, 15.2) | 10.0 (5.6, 14.3) | 6.5 (3.5,9.5) |
| 22 | 14.1 (10.9,17.3) | 9.9 (7.5 12.2) | 11.9 (10.1,13.6) | 16.5 (11.6, 21.3) | 10.3 (8.0, 12.5) | 11.6 (6.9,16.3) |
| 23 | 15.8 (13.6,18.0) | 9.9 (8.3, 11.4) | 13.4 (11.2,15.7) | 16.5 (12.2, 20.8) | 14.5 (8.9, 20.1) | 11.6 (7.1,16.2) |
| 24 | 22.3 (15.8,28.7) | 11.8 (9.7, 13.9) | 15.5 (10.8,20.1) | 23.8 (16.2, 31.4) | 15.7 (12.3,19.1) | 13.2 (8.1,18.2) |
| 25 | 22.6 (19.2,26.0) | 14.3 (11.4, 17.1) | 15.8 (12.9,18.6) | 25.0 (20.4, 29.7) | 21.6 (15.7,27.6) | 14.3 (8.8,19.8) |
| 26 | 24.4 (20.7,28.1) | 14.3 (12.2, 16.3) | N/A | 27.7 (23.0, 32.4) | 21.6 (16.0,27.3) | N/A |
| 27 | 26.5 (20.9,32.1) | 18.9 (10.8,27.0) | 21.4 (11.3,31.4) | 33.3 (20.9,45.7) | N/A | N/A |
| **Initiation of past 30-day cigarette use** | | | | | | |
| 18 | 0.0 | 0.0 | 0.0 | 0.0 | 0.0 | 0.0 |
| 19 | 2.9 (1.5, 4.4) | 2.6 (2.0,3.3) | 4.0 (2.9, 5.0) | 4.1 (1.3, 6.8) | 4.5 (2.6,6.4) | 2.2 (0.7, 3.6) |
| 20 | 5.8 (3.5, 8.2) | 4.2 (3.4,5.1) | 4.0 (2.5, 5.4) | 7.5 (4.7, 10.3) | 6.3 (3.4,9.1) | 3.3 (1.2,5.5) |
| 21 | 7.4 (3.0, 11.8) | 4.2 (3.4, 5.1) | 5.4 (3.2,7.6) | 7.5 (4.2,10.8) | 7.6 (3.4, 11.8) | 3.3 (0.0, 13.7) |
| 22 | 10.6 (7.3, 13.9) | 5.7 (3.4, 8.0) | 7.3 (4.5, 10.1) | 10.2 (6.7, 13.8) | 9.6 (7.0, 12.3) | 6.8 (3.0,10.6) |
| 23 | 13.1 (10.8, 15.5) | 6.9 (5.4, 8.3) | 8.7 (6.9,10.5) | 13.5 (9.2, 17.7) | 11.8 (7.6, 16.0) | 6.8 (3.0,10.6) |
| 24 | 14.3 (11.6, 17.0) | 7.5 (5.6, 9.4) | 8.7 (7.0, 10.3) | 16.8 (12.4, 21.4) | 12.8 (9.8, 15.7) | 8.3 (4.2,12.4) |
| 25 | 15.8 (11.2, 20.5) | 9.3 (7.0,11.6) | 11.7 (7.4,15.9) | 18.9 (13.7, 24.2) | 16.5 (11.6,21.4) | 8.9 (4.6,13.3) |
| 26 | 17.9 (14.4,21.4) | 9.3 (7.3,11.3) | 11.7 (8.4,15.0) | 18.9 (15.2, 22.7) | 16.5 (11.6,21.4) | N/A |
| 27 | N/A | 13.0 (5.7,20.2) | 16.4 (6.2,26.7) | N/A | N/A | N/A |
| **Initiation of fairly regular cigarette use** | | | | | | |
| 18 | 0.0 | 0.0 | 0.0 | 0.0 | 0.0 | 0.0 |
| 19 | 0.6 (0.0, 1.4) | 0.6 (0.3,0.9) | 1.3 (0.5,2.0) | 0.8 (0.3,1.4) | 0.4 (0.0,1.0) | 0.2 (0.0,0.5) |
| 20 | 1.5 (0.5,2.5) | 1.1 (0.5,1.6) | 1.9 (0.8,2.9) | 1.3 (0.4,2.2) | 1.0 (0.2,1.9) | 0.6 (0.1,1.2) |
| 21 | 2.7 (1.4,3.9) | 1.1 (0.5,1.6) | 2.2 (1.3,3.0) | 2.7 (1.3,4.1) | 1.0 (0.2,1.9) | 0.7 (0.1,1.4) |
| 22 | 3.1 (2.0,4.2) | 1.3 (0.6,2.1) | 2.7 (1.5,3.9) | 2.7[b] (1.3,4.1) | 1.6 (0.4,2.7) | 0.7 (0.1,1.3) |
| 23 | 3.4 (2.3,4.5) | 2.0 (1.0,3.0) | 3.0 (1.9,4.0) | N/A | N/A | N/A |
| 24 | 3.4 (2.3, 4.5) | 2.3 (1.3,3.3) | 4.1 (2.3,5.8) | 3.0 (1.5,4.5) | N/A | N/A |
| 25 | 5.6 (2.9, 8.3) | 3.1[c] (1.6,4.5) | 4.2 (2.2,6.3) | 4.5[d] (2.0,7.0) | N/A | 2.2 (0.2,4.1) |
| 26 | 5.6 (3.4, 7.8) | 3.1 (1.6,4.5) | 4.2 (2.0,6.5) | 4.5 (1.8, 7.3) | 4.6 (1.1, 8.0) | N/A |
| 27 | N/A | N/A | N/A | N/A | N/A | N/A |

[a]: Non-Hispanic Other include Non-Hispanic Asian, Multi-race, and etc.

[b]: There was not enough sample size to estimate the probability at 22 years old, so the probability in the table represents 22 years and 14 weeks.

[c]: There was not enough sample size to estimate the probability at 25 years old, so the probability in the table represents 25 and 13 weeks.

[d]: There was not enough sample size to estimate the probability at 25 years old, so the probability in the table represents 25 and 19 weeks.

[*] PATH restricted file received disclosure to publish: May 1, June 22, and July 23, 2020. United States Department of Health and Human Services. National Institutes of Health. National Institute on Drug Abuse, and United States Department of Health and Human Services. Food and Drug Administration. Center for Tobacco Products. Population Assessment of Tobacco and Health (PATH) Study [United States] Restricted-Use Files. ICPSR36231-v13.AnnArbor, MI: Inter-university Consortium for Political and Social Research [distributor], November 5, 2019. https://doi.org/10.3886/ICPSR36231.v13.

earlier ages of initiation of both past 30-day and fairly regular cigarette use. Most of the research on sex differences in the age of cigarette initiation have been conducted among youth [40–45], so more research is needed to determine the risk factors that predispose males to an

earlier age of cigarette initiation in young adulthood, as the risk factors that predispose male youth to an earlier age of cigarette initiation may be different for young adult males.

Previous studies have reported differences in the prevalence of cigarette initiation across racial/ethnic groups [19]. Our study furthers this work by indicating that there are differences in the age of cigarette initiation across racial/ethnic groups, and reports these findings by specific ages and cigarette use outcomes. Our study indirectly supports the "age crossover" hypothesis, in which the racial/ethnic differences exhibited in youth, namely that Non-Hispanic white youth have the highest prevalence of cigarette use, this reverses in young adulthood with other race/ethnicities reporting increased cigarette initiation [14]. In this study, we did not analyze participants who initiated cigarette use in youth, as they were excluded from our study because we wanted to measure cigarette initiation in young adults who did not initiate cigarette use before 18 years old, but we found that Non-Hispanic Black and Hispanic young adults had increased risk of an earlier age of cigarette initiation during young adulthood compared to Non-Hispanic White young adults. While our study only found that Hispanic young adults exhibited increased risk of initiating ever cigarette use at earlier ages during young adulthood compared to Non-Hispanic White young adults, a different nationally representative study of young adults 18–30 years old found all other race/ethnicity groups (Hispanic, Non-Hispanic Black, and Non-Hispanic Other young adults) have higher risk of initiating cigarette use in young adulthood and higher percentages of initiation of cigarette use compared to Non-Hispanic White young adults [39]. In addition, several previous studies have found that African American young adults have increased cigarette initiation in young adulthood compared to Non-Hispanic White young adults [13–15,46], which is similar to our finding that Non-Hispanic Black young adults have increased risk of initiating past 30-day cigarette use at earlier ages during young adulthood compared to Non-Hispanic White young adults. The increased risk for initiating cigarette use among Non-Hispanic Black young adults could be explained by Minority Diminished Return theory [47,48], which posits that the factors known to be protective against cigarette use in Non-Hispanic White young adults are less protective among Non-Hispanic Black young adults. Observed racial inequalities in the education system, labor market, and other institutions contribute to these diminished protective effects [48–52]. For example, the educational system has been shown to discriminate against African Americans, reducing the gain that comes with higher education [50]. The labor market discriminates against African Americans in hiring and wages [53], reducing the protection that comes with full-time work [52]. These explanations are extremely important to consider in future research in order to combat the health effects associated with institutionalized racism [48]. In terms of the current study, an earlier age of cigarette initiation during young adulthood among Non-Hispanic Black young adults increases the number of years of cigarette use, which increases the risk of adverse health consequences [1,18]. Other possible explanations include predatory marketing by the tobacco industry in low socioeconomic status (SES) and high minority neighborhoods, flavor branding, SES, quality of healthcare, among others [19,48]. The current study findings highlight a need for future studies to examine potential risk and protective drivers of tobacco use disparities exist across these groups.

There are several publications that have documented previous e-cigarette use as a risk factor for subsequent cigarette initiation among young adults [22,54,55]. A previous study of PATH wave 1 (2013–2014) adult (18+) never cigarette users found that e-cigarette use at wave 1 was associated with ever (AOR = 2.9; 95%CI = 2.0–4.0) and past 30-day (AOR = 3.2; 95%CI = 2.1–4.9) cigarette use one year later after controlling for sociodemographic characteristics and other tobacco product use [11]. Our study is consistent with this previous study and extends findings specifically to young adults (18–24 years old); namely, that previous e-cigarette use increases the risk of an earlier age of ever, past 30-day and fairly regular cigarette initiation

during young adulthood by 48%, 71%, and 119%, respectively. The importance of our findings should be considered because while e-cigarettes have been marketed as a cessation tool, our findings show that e-cigarettes may predispose young adults to initiate cigarette use at earlier ages than never e-cigarette users. Similarly, our results are consistent with research that has found that previous use of other tobacco products is associated with subsequent cigarette use in adults [11], college students [56,57], and youth [58–60]. Our study again extends these findings specifically to young adults. A recent PATH study found that past 30-day poly-tobacco cigarette use (use of cigarettes and at least 1 other tobacco product in the past 30-days) is common among young adults (65.2% of tobacco users) at wave 1, and continued poly-tobacco cigarette use was the most persistent pattern of tobacco across waves 2 and 3 (37.1%) [61].

While there are many prevention interventions aimed at youth [62–64], young adults are more often targeted with cessation programs [62,63]. However, given that we found that cigarette initiation does occur in young adulthood, our study shows the necessity of cigarette interventions to prevent cigarette use, especially aimed at young adults 18–21 years old. The results from our study suggest that the focus of the interventions should be tailored towards males and ethnic minority groups (especially Hispanic and Non-Hispanic Black young adults) between 18–21 years old. Given the concerning findings regarding tobacco-related health disparities in ethnic minority groups [19], it is possible that these young adults could benefit from culturally-relevant prevention interventions targeted specifically towards them. Previous research has identified culturally-relevant strategies, such as messages that highlight the impact of tobacco industry marketing and advertising on cigarette use behaviors in minorities, telling success stories of tobacco quitters, emphasizing the success of community leaders who are non-users, and messages that include historical lessons on tobacco use could be particularly received by the African American community [65,66]. FDA has implemented a new cigarette prevention campaign that includes modifying tobacco prevention and intervention campaigns to other languages, messages that associated living tobacco-free with a hip-hop lifestyle, and including multicultural cigarette-free role-models, which may help to prevent cigarette initiation in multicultural young adults [67,68]. In addition, population-based strategies, such as FDA's required cigarette health warnings that include text, color graphics, and descriptions of some of the less-known but serious health risks from cigarette use [69,70], will be paramount to reduce cigarette initiation overall. Other effective strategies that can be implemented at the population level include mass media campaigns [66,71], comprehensive smoke-free policies [72], state-funded cessation programs [73,74], reducing the nicotine content in cigarettes [75], menthol bans [76], and increasing the tax on cigarettes [77]. Additional research is needed to better inform the development of culturally-relevant tobacco prevention interventions for young adults, and to evaluate the reach and coverage of those interventions.

Finally, our study can be used as a baseline for any future evaluation of the effectiveness of the Tobacco 21 law, a U.S. federal law that changed the minimum age of tobacco sale from 18 years old to 21 years old in December 2020, to determine if this law has shifted the distribution of the age of initiation of cigarette use by preventing lawful access to cigarettes [20]. The findings presented in this study can also be used by future interventions to establish targets for reductions in cigarette use by specific ages. For example, if a prevention intervention is targets young adults 21 years and younger who had never used cigarettes by 18 years old, we would expect that ever cigarette initiation would be less than 10.6% if the intervention was effective beyond current interventions. However, it should be noted that 19 states and Washington D. C. had Tobacco 21 laws that were implemented prior to the federal law [78]. This study's findings adds to the body of evidence that supports more stringent regulations by the Food and Drug Administration (FDA) on cigarette product advertising and marketing (e.g., new graphic

health warning label), culturally-relevant interventions, and population-based interventions [79].

## Strengths and limitations

We use nationally representative data to prospectively estimate the age of initiation of ever cigarette use, past 30-day cigarette use, and fairly regular cigarette use, which is one of the strengths of our study. With PATH data, we were able to follow a nationally-representative sample of participants longitudinally over multiple years (2013–2017). The use of interval censoring using non-parametric methods to estimate the age of initiation prospectively is another strength, as our results do not depend on parametric model assumptions. PATH participant birth dates are not included in the restricted-use data due to participant confidentiality, which prevented us from obtaining participants' exact age at each wave, and is a limitation. However, we overcome this limitation by using the number of weeks between survey waves and interval-censoring to estimate the age of initiation. An additional limitation is that the precision of our estimates was diminished when estimates are provided by race/ethnicity for each age due to reduced sample size, but the confidence intervals are provided to allow readers to account for the variability of the estimates. Another limitation is that while PATH participants were followed-up from 2013–2017, tobacco product use behaviors may have changed among young adults since 2017, including combustible tobacco cigarette use, which could have an impact the age of initiation of cigarette use behaviors. Finally, participants were not asked the exact date that they initiated cigarette use. However, this may be unrealistic for participants to accurately remember and thus would be subject to recall bias.

## Conclusions

In conclusion, this study provides strong evidence that a substantial portion of contemporary cigarette initiation does occur in young adulthood and gives estimates for the age of cigarette initiation. Among 18 year olds who have never smoked a cigarette, 22.7% report initiating combustible tobacco cigarettes use by the age of 27. In addition, males, Hispanic young adults, Non-Hispanic Black young adults, previous e-cigarette users and those who have used other tobacco products before cigarette initiation have an increased risk of initiating cigarette use at earlier ages during young adulthood. Based on these findings, cigarette prevention and education campaigns should be expanded to target young adults, especially 18–21 year olds.

## Supporting information

**S1 Table. Total number of other tobacco products, excluding e-cigarettes, ever used prior to the initiation of cigarette use outcomes among PATH young adult (ages 18–24 years old) never cigarette users at their first wave of adult study participation, 2013–2016.** (DOCX)

## Acknowledgments

We thank the reviewers of the previous version of the manuscript for helping us to improve this final version.

## Author Contributions

**Conceptualization:** Adriana Pérez, Baojiang Chen, Cheryl L. Perry, Kymberly L. Sterling, Melissa B. Harrell.

**Formal analysis:** Adriana Pérez, Arnold E. Kuk, Elena Penedo, Roi San N'hpang, Baojiang Chen.

**Funding acquisition:** Adriana Pérez, Meagan A. Bluestein, Baojiang Chen, Cheryl L. Perry, Melissa B. Harrell.

**Investigation:** Adriana Pérez, Elena Penedo, Roi San N'hpang.

**Methodology:** Adriana Pérez.

**Project administration:** Adriana Pérez.

**Resources:** Adriana Pérez.

**Supervision:** Adriana Pérez, Meagan A. Bluestein.

**Visualization:** Adriana Pérez, Roi San N'hpang, Baojiang Chen.

**Writing – original draft:** Adriana Pérez, Roi San N'hpang.

**Writing – review & editing:** Adriana Pérez, Arnold E. Kuk, Meagan A. Bluestein, Elena Penedo, Baojiang Chen, Cheryl L. Perry, Kymberly L. Sterling, Melissa B. Harrell.

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
