## [Decision Letter · Decision Letter 0]

11 Jan 2021

PONE-D-20-33267

Prospective estimation of the age of initiation of cigarettes among young adults (18-24 years old): findings from the Population Assessment of Tobacco and Health (PATH) waves 1-4 (2013-2017)

PLOS ONE

Dear Dr. Pérez,

Thank you for submitting your manuscript to PLOS ONE. After careful consideration, we feel that it has merit but does not fully meet PLOS ONE’s publication criteria as it currently stands. Therefore, we invite you to submit a revised version of the manuscript that addresses the points raised during the review process.

We look forward to receiving your revised manuscript.

Kind regards,

Stanton A. Glantz

Academic Editor

PLOS ONE

Journal Requirements:

2. In your statistical analyses, please state whether you accounted for repeated measurements and the potential effect of clustering by state. For example, did you consider using multilevel models?

Reviewers' comments:

Reviewer's Responses to Questions

**Comments to the Author**

1. Is the manuscript technically sound, and do the data support the conclusions?

Reviewer #1: Partly

Reviewer #2: Yes

2. Has the statistical analysis been performed appropriately and rigorously? 

Reviewer #1: Yes

Reviewer #2: Yes

3. Have the authors made all data underlying the findings in their manuscript fully available?

Reviewer #1: Yes

Reviewer #2: Yes

4. Is the manuscript presented in an intelligible fashion and written in standard English?

Reviewer #1: Yes

Reviewer #2: Yes

5. Review Comments to the Author

Reviewer #1: COMMENTS TO AUTHORS

The manuscript entitled “Prospective estimation of the age of initiation of cigarettes among young adults (18-24 years old): findings from the Population Assessment of Tobacco and Health (PATH) waves 1-4 (2013-2017)”

addresses the important topic of late initiation of cigarette smoking, at ages 18-24 which is past the ages when the vast majority of smokers begin smoking. Thus, this broad topic has potentially important public health implications.

A great strength is that this research is embedded within the PATH Study, a very rich nationally representative resource with longitudinal data to assess this question. Further, the research team seemed to have excellent expertise in the data management and statistical analytic approaches to maximize the use of these detailed data. An interesting finding was that whereas individuals who identify as non-Hispanic White are much more likely to start smoking at younger ages compared with other racial/ethnic groups, this seems to be reversed during the 18-24 year-old age period.

Despite these strengths, many issues described below were identified that need to be addressed to strengthen the manuscript.

Major Comments

1. It was a negative beginning to have an abstract with no scientific rationale or background to provide context, creating the impression from the outset that the quantitative expertise is not balanced by substantive expertise.

2. The very first sentence of the manuscript needs work. With cigarette smoking, providing a partial list of smoking-caused diseases is not informative enough and referring to cigarette smoking as “…one of the associated factors of…death in the United States” grossly understates its importance as the leading cause of premature death estimated to cause 480,000 deaths per year. Further, cigarette smoking has not been found to be a cause of breast cancer.

3. In the Introduction, in the first sentence of the second paragraph a seemingly identical publication using PATH Study data (but with shorter follow-up) is referenced that includes members of the same research team as this manuscript. This reduced enthusiasm about the novelty of this research, making it seem like little advance over the already published work (reference 9).

4. In the current complex tobacco marketplace precision in identifying products assumes increased importance. It is helpful to define terms early on, to make clear to the reader that when the manuscript refers to “cigarettes” this is referring to “combustible tobacco cigarettes” (or something of the sort).

5. It was puzzling that the manuscript seemed to be focused on 18-24 year-olds from the title onward, but then all of the sudden data is presented about 27 and 28 year-olds. Presumably, this is due to the 23-24 year-olds at baseline being followed over time, but this age issue needs to be clarified early on and handled uniformly throughout the manuscript.

6. Following up on Comment #5, the biggest increase occurring between 18 and 19 years-old is completely expected and makes sense, but how could the second biggest increase in past 30-day use be among 27 to 28 years-old? This does not adhere to prior expectation and thus deserves more attention.

7. In Table 2 the terminology “hazard functions” is used over the more accessible and public health relevant term “cumulative incidence” (which is relegated to a footnote). Changing this will improve clarity for readers from different disciplines.

8. An extremely important variable to consider in this research is prior tobacco product use. The authors accounted for this by creating a variable that summed up the number of products used in the past (0, 1, and >=2). While this has relevance, a question of burning scientific interest is the relationship between prior use of a specific tobacco product and the risk of future smoking—that specific tobacco product is e-cigarettes. So, this was a major missed opportunity to document the association between prior e-cigarette use in relation to future initiation of cigarette smoking in young adulthood. Even so, the importance of prior tobacco product use is clear by the very strong associations between prior tobacco product use and future cigarette smoking. A p-value for trend should be included for this variable (likely highly statistically significant in every model). Remarkably, this strong association was not mentioned in the Discussion section when it should have been a major focus. Overall, the handling of other tobacco product use was a missed opportunity and this was rated as a major weakness.

9. In the Discussion section, the text in lines 363 to 367 attempts to delve into disparities in lung cancer which the authors eventually state is beyond the scope of this research—this text should be omitted from the manuscript for this very reason.

10. The manuscript ended abruptly with “Strengths and Limitations” without a synopsis of the findings and conclusions about the results.

Minor Comments

1. In the Methods section, line 115-117 there is a statement “…this resulted in 5,523 (N=19,548,811) young adult never cigarette users…” It was unclear what

“N” was when this appeared. It becomes clearer later that this “N” stands for the weighted frequency, but this should be clarified when it is first introduced.

2. Please mention in Table 1 that the capital N is the weighted frequency in the row immediately below the row titled “Never cigarette users at first wave of adult study participation”.

3. It is very unusual in my peer review experience to have tables embedded directly in the text, and this formatting makes it difficult to read and review the manuscript especially when the tables were broken up across multiple pages.

4. In lines 181-184 the reader is provided with the information that “Each hazard function per outcome took approximately 30 hours to complete using the restricted-use data server [23], with an additional 30 hours for sex and 30 hours for race/ethnicity, resulting in a total run time of approximately 360 hours (=90 hours*4 outcomes) to complete all analyses.” This sentence is not necessary, and the manuscript would be strengthened by omitting it.

Reviewer #2: This manuscript presents the findings from an analysis of longitudinal PATH data to assess the age of cigarette initiation among young adults aged 18-24 who were never cigarette smokers at baseline. The study found that among young adults who never smoked at baseline, the greatest increase in initiation occurred between 18-19 years old, and by age 21, approximately 1 in 10 had initiated ever cigarette smoking and about 1 in 13 had initiated past 30 day smoking. By demographics, initiation was more likely among males, Hispanics, and non-Hispanic Blacks.

Overall, this study is methodologically sound, and would have important implications for informing tobacco prevention strategies among young adults, including regulatory actions by the US Food and Drug Administration. Nonetheless, it could still be improved with some relatively straightforward revisions described below.

1. Page 2. Abstract. Conclusion, First sentence. The primary call to action could be stated more explicitly (e.g. “The prominent initiation of cigarette smoking among young adults reinforces the need for prevention strategies among this population”).

2. Page 4. Introduction. First paragraph. Cigarette smoking has only been declining since the mid 1960s among adults. Youth smoking rates increased prominently until the 1998 Master Settlement Agreement, and have been declining since.

3. Page 5. Introduction. Last Paragraph. It would be helpful if the authors explicitly articulated the inherent novelty of the present study and how it builds about the existing scientific literature. There are references to previous studies, but the Introduction section doesn’t make a clear connection as to how the study specifically fills an existing void. It’s also important that the authors cite several recent PATH studies published on this issue, which aren’t otherwise included. For example, earlier this year Tobacco Control published a special supplement using longitudinal analyses from PATH during 2013-2016, including Stanton et al (initiation among youth, young adults, and adults), and Taylor et al (exclusive and polytobacco cigarette use among youth, young adults, and adults).

4. Page 6. Study Design and Participants. Given that many readers will not have a nuanced understanding of PATH methods, it would be beneficial if the authors better clarified when waves occurred – e.g. Wave 1 (2013-2014), Wave 2 (2014-2015), etc. It’s also recommended that they use consistent terminology when referencing these periods. For example, in some areas of the manuscript “Wave X” is used, while elsewhere the years of data collection are used, to reference the findings for that period.

5. Page 6. Study Design and Participants. The breakdown of study participants across waves, including excluded participants, is a bit difficult to follow in narrative form. It’s recommended that the authors add a figure with a flow chart that visually clarifies how the final sample was ultimately derived.

6. Page 7. Cigarette Use Measures. It’s not clear what is meant by “derived variable” for the ever use measure. It’s recommended that more clarity be provided on the precise framing of the question(s) that were used to create this measure.

7. Page 7. Other Tobacco Product Use. It’s not clear why the authors only examined ever tobacco product use at participants’ first PATH wave of participation. Couldn’t initiation have also occurred during a subsequent wave (e.g. wave 2), prior to assessment over time (e.g. wave 4)? Further clarity on this analytic choice is warranted to better justify the approach to readers.

8. Page 10. Table 1. The weighted percentages in the table are confusing, particularly having missing values in excess of 8,000 for the sex category. It would be much more straightforward for the reader if the authors included unweighted sample sizes in this table (or both unweighted and weighted if the authors feel strongly).

9. Page 17. Discussion. First paragraph. The authors reference a “past 30 day” estimate from the National Health Interview Survey (NHIS), but current use from that survey is usually reported using a 100 lifetime threshold and “everyday”/”someday” use. Please cross-check the actual reference to ensure that the language you’ve used is accurate as framed.

10. Pages 17-18. Discussion. The authors nicely describe their findings in comparison to other studies, including the variations by sex and race/ethnicity. However, the utility and relevance of this manuscript would be greatly enhanced if the authors explained why these disparities are likely occurring, including potential drivers that are creating these inequities.

11. Page 18. Discussion. The authors appropriately call for prevention interventions among young adults, but it could be more clearly stated in the narrative (see comment #1 above). Additionally, it would be immeasurably more helpful if the authors explained what interventions they’re talking about. The most effective prevention measures are population-based policies, of which they are many, including price, smoke-free, tobacco 21, and even product restrictions (e.g. flavors). At present, there’s a lot of talk about FDA regulation, which is fine – but states and communities are not preempted by the Family Smoking Prevention and Tobacco Control Act, and indeed much of the momentum on population-based policies are presently occurring at the state and local level. Not acknowledging any of the strategies that can move the needle forward on preventing young adult use at the national, state, and local levels is really a missed opportunity here.

12. Page 19. Discussion. The authors call for culturally relevant prevention interventions, which is reasonable. But they also need to reinforce that equitable coverage of interventions is needed in the first place. Ultimately, being covered by an intervention in the first place is going to be a more critical line of defense when it comes to prevention than a targeted intervention.

13. Page 19. The authors reference the “Tobacco 21 Law”, but this won’t be intuitive to lay readers. There needs to be more context around this statement, which presumably references the Federal law adopted in December 2019 to increase the age of sale nationally to 21. That said, it would also seem relevant to reference the numerous state and local tobacco 21 laws that were also implemented prior to the national law. Also, it’s not clear why the authors have just focused on advertising and marketing at the end of this paragraph. There are numerous regulatory actions related to manufacturing, marketing, and sale that could be taken by FDA to mitigate young adult initiation besides just addressing advertising and marketing (e.g. menthol, nicotine reduction, etc).

14. Page 19. The limitations section could be more robustly framed. In addition to the fact that age of initiation had to be derived using a proxy approach, there are also important implications specifically related to the longitudinal nature of the data, including loss to attrition over time and a relatively small sample that prevented more nuanced precision around certain assessed variables (e.g. race/ethnicity). It’s also important to note the age of the data. These data are from nearly a half decade ago, and there’s been considerable volatility in the tobacco product marketplace in recent years, including increasing e-cigarette use among young adults and enhanced adoption of various population-level interventions at the national/state/local levels – all of which could impact the extent of these observed findings from 2013-2016.

15. It’s recommend that the authors add a clear conclusion paragraph that summarizes the major take home message for the reader and provides a clear call to action. At present, it ends with the negative aspects of the manuscript (i.e. the limitations), rather than an actionable summary and public health implications statement.

6. PLOS authors have the option to publish the peer review history of their article (what does this mean?). If published, this will include your full peer review and any attached files.

Reviewer #1: No

Reviewer #2: No

---

## [Decision Letter · Decision Letter 1]

20 Apr 2021

PONE-D-20-33267R1

Prospective estimation of the age of initiation of cigarettes among young adults (18-24 years old): findings from the Population Assessment of Tobacco and Health (PATH) waves 1-4 (2013-2017)

PLOS ONE

Dear Dr. Pérez,

Thank you for submitting your manuscript to PLOS ONE. After careful consideration, we feel that it has merit but does not fully meet PLOS ONE’s publication criteria as it currently stands. Therefore, we invite you to submit a revised version of the manuscript that addresses the points raised during the review process.

We look forward to receiving your revised manuscript.

Kind regards,

Stanton A. Glantz

Academic Editor

PLOS ONE

Journal Requirements:

Reviewers' comments:

Reviewer's Responses to Questions

**Comments to the Author**

1. If the authors have adequately addressed your comments raised in a previous round of review and you feel that this manuscript is now acceptable for publication, you may indicate that here to bypass the “Comments to the Author” section, enter your conflict of interest statement in the “Confidential to Editor” section, and submit your "Accept" recommendation.

Reviewer #1: (No Response)

Reviewer #2: All comments have been addressed

2. Is the manuscript technically sound, and do the data support the conclusions?

Reviewer #1: Yes

Reviewer #2: Yes

3. Has the statistical analysis been performed appropriately and rigorously? 

Reviewer #1: Yes

Reviewer #2: Yes

4. Have the authors made all data underlying the findings in their manuscript fully available?

Reviewer #1: Yes

Reviewer #2: Yes

5. Is the manuscript presented in an intelligible fashion and written in standard English?

Reviewer #1: Yes

Reviewer #2: Yes

6. Review Comments to the Author

Reviewer #1: The authors have been very responsive to the initial round of critiques, and the overall quality of the manuscript has improved significantly.

Now that the concerns raised on the prior review have been addressed, only a few minor comments emerge that can be readily addressed.

Comment 1. Given the events of the past year and enhanced focused on anti-racism, which the authors are clearly quite knowledgeable about given the excellent new text on this issue in the Discussion section, the manuscript text referring to racial/ethnic groups most often uses a dehumanizing approach that probably would not be corrected by a technical editor. This is when language refers solely to "Hispanics" or "Non-Hispanic Blacks" or "Non-Hispanic Others or "Non-Hispanic Whites." Preferred language occurs elsewhere when the authors refer to "Hispanic young adults" etc. because this language acknowledges the humanity of the individuals in these groups. Other strategies include terms such as "Hispanic Americans" or "Hispanic people" etc. Consistent use of this more humanizing terminology is essential.

Comment 2. The Discussion section is now excellent. The authors demonstrate superb insight in text that addresses the racial/ethnic differences in initiation in childhood and adolescent compared with the results of the present study of young adults. However, given the potential confusion that may arise from the text elsewhere in the discussion section particularly before the key text referred to in the prior sentence, it is critical to avoid confusion. For example, the sentence starting on line 374 will be more precise to say something like "...more likely to initiate ever cigarette use at an earlier age OF YOUNG ADULTHOOD than females..." The main point is in these key sentences in the Discussion to clearly specify the age range studied so these statements are not taken out of context by the cursory reader to mean overall ages of initiation rather than age of initiation in young adulthood.

Comment 3. Discussion section, line 443 "This finding is concerning given the increased health risk associated with using more than 1 tobacco product." This sentence as is should be taken out. The fact is that if the one tobacco product used is combustible tobacco cigarettes, that is the maximal risk. If a tobacco user smokes combustible tobacco cigarettes and e-cigarettes, and the e-cigarettes result in fewer combustible tobacco cigarettes this could result in reduced risk. So this is complicated and best avoided. The paragraph ends nicely with the factual statements and no additional concluding sentence is needed.

Comment 4. Discussion section, line 504: The authors refer to "conventional" cigarettes. This is more subjective that "combustible tobacco cigarettes" because the word "conventional" carries subjective meaning that implies "usual" or "normal". So please change to "combustible tobacco" for this reason.

Reviewer #2: The authors have adequately addressed my original comments. The only exception is original comment #14. Although I can appreciate that more recent data aren't available, which isn't particularly surprising given the glacial speed with which FDA has made data from PATH publicly available, this doesn't negate the fact that the data are old. The author are correct in stating that they aren't a half decade old - a sizable portion of it is actually more than a half decade old. Although the most recent data pull is from 2017, the baseline estimates among never smokers were pulled from data as early as 2013-2014, which was a very different time in terms of both policy and usage behaviors. The tobacco product landscape has changed considerably since 2013-2014, including since 2017. Acknowledging that these recent changes could impact more recent initiation behaviors is perfectly within the realm of possibilities and would seem like something the authors could easily insert into the limitations. For example, something like: "Respondents were followed-up as recently as 2017; however, tobacco product use behaviors have changed among youth in more recent years, including for cigarette smoking, which could have an impact more recent initiation behaviors."

7. PLOS authors have the option to publish the peer review history of their article (what does this mean?). If published, this will include your full peer review and any attached files.

Reviewer #1: No

Reviewer #2: No

---

## [Author Response · Author response to Decision Letter 1]

20 Apr 2021

We greatly appreciate the previous feedback from the editor and reviewers, which gave us the opportunity to improve the manuscript and make it stronger. 

Journal Requirements:

Response: We have reviewed all references and found that no papers have been retracted. We have reviewed all references and found no mistakes that need to be corrected. 

Reviewer 1

Comment 1. Given the events of the past year and enhanced focused on anti-racism, which the authors are clearly quite knowledgeable about given the excellent new text on this issue in the Discussion section, the manuscript text referring to racial/ethnic groups most often uses a dehumanizing approach that probably would not be corrected by a technical editor. This is when language refers solely to "Hispanics" or "Non-Hispanic Blacks" or "Non-Hispanic Others or "Non-Hispanic Whites." Preferred language occurs elsewhere when the authors refer to "Hispanic young adults" etc. because this language acknowledges the humanity of the individuals in these groups. Other strategies include terms such as "Hispanic Americans" or "Hispanic people" etc. Consistent use of this more humanizing terminology is essential.

Response: We agree 100% with the sentiment behind this feedback. We initially left out the words “young adults” because of trying to save on word count, and we appreciate the opportunity to refer to our participants in a humanistic way. Throughout the abstract and entire manuscript, we have updated our text to refer to our participants as “[race] young adults”. 

Comment 2. The Discussion section is now excellent. The authors demonstrate superb insight in text that addresses the racial/ethnic differences in initiation in childhood and adolescent compared with the results of the present study of young adults. However, given the potential confusion that may arise from the text elsewhere in the discussion section particularly before the key text referred to in the prior sentence, it is critical to avoid confusion. For example, the sentence starting on line 374 will be more precise to say something like "...more likely to initiate ever cigarette use at an earlier age OF YOUNG ADULTHOOD than females..." The main point is in these key sentences in the Discussion to clearly specify the age range studied so these statements are not taken out of context by the cursory reader to mean overall ages of initiation rather than age of initiation in young adulthood.

Response: We appreciate this piece of feedback and have made the change in this instance, as well as several other places in the discussion. 

Comment 3. Discussion section, line 443 "This finding is concerning given the increased health risk associated with using more than 1 tobacco product." This sentence as is should be taken out. The fact is that if the one tobacco product used is combustible tobacco cigarettes, that is the maximal risk. If a tobacco user smokes combustible tobacco cigarettes and e-cigarettes, and the e-cigarettes result in fewer combustible tobacco cigarettes this could result in reduced risk. So this is complicated and best avoided. The paragraph ends nicely with the factual statements and no additional concluding sentence is needed.

Response: We have removed this sentence from the updated version of the manuscript. 

Comment 4. Discussion section, line 504: The authors refer to "conventional" cigarettes. This is more subjective that "combustible tobacco cigarettes" because the word "conventional" carries subjective meaning that implies "usual" or "normal". So please change to "combustible tobacco" for this reason.

Response: We appreciate this nuance in language and as tobacco health researchers, we would never want to normalize tobacco use behaviors. We have updated this sentence to say “combustible tobacco cigarettes”. 

Reviewer #2:

Comment 1: The authors have adequately addressed my original comments. The only exception is original comment #14. Although I can appreciate that more recent data aren't available, which isn't particularly surprising given the glacial speed with which FDA has made data from PATH publicly available, this doesn't negate the fact that the data are old. The author are correct in stating that they aren't a half decade old - a sizable portion of it is actually more than a half decade old. Although the most recent data pull is from 2017, the baseline estimates among never smokers were pulled from data as early as 2013-2014, which was a very different time in terms of both policy and usage behaviors. The tobacco product landscape has changed considerably since 2013-2014, including since 2017. Acknowledging that these recent changes could impact more recent initiation behaviors is perfectly within the realm of possibilities and would seem like something the authors could easily insert into the limitations. For example, something like: "Respondents were followed-up as recently as 2017; however, tobacco product use behaviors have changed among youth in more recent years, including for cigarette smoking, which could have an impact more recent initiation behaviors."

Response: We appreciate this perspective from reviewer 2 and agree that this can be considered a limitation. We have included a paraphrased version of this sentence as a limitation in the strengths and limitations section.

---

## [Editor Report · Decision Letter 2]

23 Apr 2021

Prospective estimation of the age of initiation of cigarettes among young adults (18-24 years old): findings from the Population Assessment of Tobacco and Health (PATH) waves 1-4 (2013-2017)

PONE-D-20-33267R2

Dear Dr. Pérez,

We’re pleased to inform you that your manuscript has been judged scientifically suitable for publication and will be formally accepted for publication once it meets all outstanding technical requirements.

Kind regards,

Stanton A. Glantz

Academic Editor

PLOS ONE
---

## [Editor Report · Acceptance letter]

27 Apr 2021

PONE-D-20-33267R2 

Prospective estimation of the age of initiation of cigarettes among young adults (18-24 years old): findings from the Population Assessment of Tobacco and Health (PATH) waves 1-4 (2013-2017) 

Dear Dr. Pérez:

I'm pleased to inform you that your manuscript has been deemed suitable for publication in PLOS ONE. Congratulations! Your manuscript is now with our production department. 

Kind regards, 

on behalf of

Professor Stanton A. Glantz 

Academic Editor

PLOS ONE